# Resonance-Enhanced Detection of Metals in Aerosols using Single Particle Mass Spectrometry

Johannes Passig[1,2,3], Julian Schade[2,3], Ellen Iva Rosewig[2,3], Robert Irsig[3,4], Thomas Kröger-Badge[2,3], Hendryk Czech[1,2,3], Martin Sklorz[1], Thorsten Streibel[1,2], Lei Li[5], Xue Li[5], Zhen Zhou[5], Henrik Fallgren[6], Jana Moldanova[6], and Ralf Zimmermann[1,2,3]

[1]Joint Mass Spectrometry Centre, Cooperation Group 'Comprehensive Molecular Analytics' (CMA), Helmholtz Zentrum München, 85764 Neuherberg, Germany
[2]Joint Mass Spectrometry Centre, Chair of Analytical Chemistry, University Rostock, 18059 Rostock, Germany
[3]Department Life, Light & Matter, University of Rostock, 18051 Rostock, Germany
[4]Photonion GmbH, 19061 Schwerin, Germany
[5]Institute of Mass Spectrometry and Atmospheric Environment, Jinan University, Guangzhou 510632, China and Guangzhou Hexin Instrument Co., LTD, Guangzhou 510530, China
[6]IVL Swedish Environmental Research Institute, 411 33 Gothenburg, Sweden

*Correspondence to:* Johannes Passig (johannes.passig@uni-rostock.de)


**Abstract.** We describe resonance effects in laser desorption/ionization (LDI) of particles that substantially increase the
sensitivity and selectivity to metals in single particle mass spectrometry (SPMS). Within the proposed scenario, resonant
light absorption by ablated metal atoms increases their ionization rate within a single laser pulse. By choosing the
appropriate laser wavelength, the key micronutrients Fe, Zn and Mn can be detected on individual aerosol particles with
considerably improved efficiency. These ionization enhancements for metals apply to natural dust and anthropogenic
aerosols, both important sources of bioavailable metals to marine environments. Transferring the results into applications, we
show that the spectrum of our KrF-excimer laser is in resonance with a major absorption line of iron atoms. To estimate the
impact of resonant LDI on the metal detection efficiency in SPMS applications, we performed a field experiment on ambient
air with two alternately firing excimer lasers of different wavelengths. Herein, resonant LDI with the KrF-excimer laser
(248.3 nm) revealed iron signatures for many more particles of the same aerosol ensemble compared to the more common
ArF-excimer laser line of 193.3 nm (non-resonant LDI of iron). Many of the particles that showed iron contents upon
resonant LDI were mixtures of sea salt and organic carbon. For non-resonant ionization, iron was exclusively detected in
particles with a soot contribution. This suggests that resonant LDI allows a more universal and secure metal detection in
SPMS. Moreover, our field study indicates relevant atmospheric iron transport by mixed organic particles, a pathway that
might be underestimated in SPMS measurements based on non-resonant LDI. Our findings show a way to improve the
detection and source attribution capabilities of SPMS for particle-bound metals, a health-relevant aerosol component and an
important source of micronutrients to the surface oceans affecting marine primary productivity.

## 1 Introduction

Natural and anthropogenic aerosols play a pivotal role in global climate and biogeochemical cycles, yet limited ambient observations result in large uncertainties. While sulfate and carbonaceous aerosols are intensively investigated for their climate effects (Wang et al., 2016; Seinfeld and Pandis, 2016; Kanakidou et al., 2005; Bond et al., 2013; Sofiev et al., 2018), the particle-bound metals have far-reaching impacts on ecosystems and human health. The redox cycling activity of inhaled transition metals such as iron (Fe) induces oxidative stress and is involved in severe health effects from air pollution (Ye et al., 2018; Oakes et al., 2012; Fang et al., 2017). Furthermore, atmospheric particles are important sources of marine micronutrients (Mahowald et al., 2018; Jickells et al., 2005). The highly soluble, and thus more bioavailable Fe from anthropogenic aerosols that adds to the larger flux of rather insoluble mineral dust is assumed to affect primary production and carbon export in a significant part of the world's oceans (Ito and Shi, 2016; Li et al., 2017; Ito, 2015). Beyond Fe, further biologically important trace metals exert health effects (Gaur and Agnihotri, 2019) or can modulate primary production (Mahowald et al., 2018). For example, as enzyme co-factors they can determine which enzymes cells can express, affecting the composition of microbial communities (Boyd et al., 2017). Productivity-limiting deficiencies of e.g. manganese (Mn) and zinc (Zn) have been reported for marine regions (Mahowald et al., 2018). Zinc is also associated with toxicological responses to wood combustion aerosols (Uski et al., 2015; Kanashova et al., 2018). However, the magnitude and variability of anthropogenic sources of bioavailable metals in the sea are poorly characterized. Recently, anthropogenic fluxes and sources of Fe were estimated using isotope fingerprinting (Conway et al., 2019). Such studies require measurable differences between natural and anthropogenic isotope distributions of the respective metals. Alternative methods, preferably providing detailed source information, indicating the metal's bioavailability and acquiring episodic deposition events are required to refine the global distribution models with observational data.

Several mass spectrometry based analytical techniques for aerosol characterization have been developed, with single particle mass spectrometry (SPMS) being a real-time method obtaining the size and a chemical profile from individual particles (Pratt and Prather, 2012; Laskin et al., 2018). In SPMS, the particles are introduced into a vacuum, individually sized and exposed to intense UV laser pulses that form a partly ionized plume (laser desorption/ionization, LDI) (Hinz and Spengler, 2007; Murphy, 2007). Ions are extracted and analysed with respect to their mass-to-charge ratio (m/z). Typically observed ions are e.g. organic fragments, salts, ammonia, nitrate, sulfate, alkali metals, mineral components such as silicate and carbon clusters from elemental or organic carbon (EC or OC). Along with the single-particle aspect, SPMS stands out for its metal detection capabilities that yield unique source information data (Dall'Osto et al., 2016b; Pratt and Prather, 2012; Arndt et al., 2017; Dall'Osto et al., 2016a). For example, vanadium can indicate ship emissions (Healy et al., 2009; Ault et al., 2010) and signal patterns of e.g. aluminium, silicon and calcium point on soil dust particles (Sullivan et al., 2007). However, compound-specific ionization efficiencies differ significantly. For example, the particle's humidity and its main composition can have a strong effect on the detection of particle compounds (Neubauer et al., 1998), known as matrix effects. These effects are associated with several poorly determined interactions at the particle surface and in the desorbed plume affect ion

formation (Reilly et al., 2000; Reinard and Johnston, 2008; Hinz and Spengler, 2007; Murphy, 2007; Wade et al., 2008; Hatch et al., 2014; Schoolcraft et al., 2000), reduce detection efficiencies and complicate quantification approaches (Healy et al., 2013; Gemayel et al., 2017; Gross et al., 2000; Fergenson et al., 2001; Qin et al., 2006; Zhou et al., 2016; Shen et al., 2019). These difficulties can be mitigated if the desorption and ionization are separated in a two-step process, and ions are formed in the gaseous plume as demonstrated for aromatic hydrocarbons (Morrical et al., 1998; Bente et al., 2008; Woods et al., 2001). In such a two-step approach, thermal or laser desorption (LD) is often followed by Resonance-Enhanced Multiphoton Ionization (REMPI), a gas-phase ionization technique that is highly sensitive and selective for aromatic molecules (Gunzer et al., 2019). The LD-REMPI approach yields detailed mass spectra of the health-relevant polycyclic aromatic hydrocarbons (PAHs) - ubiquitous trace compounds of combustion particles (Bente et al., 2009; Li et al., 2019; Passig et al., 2017; Schade et al., 2019). Resonant laser ablation of metals, where the leading edge of the laser pulse ablates atoms from a solid sample that are then ionized by the same pulse, have been studied some time ago for Laser Microprobe Mass Analysis (LAMMA) from surfaces (Verdun et al., 1987; McLean et al., 1990). However, to our best knowledge, such effects have so far not been recognized and applied in aerosol/single particle mass spectrometry. In the current study, we report on such wavelength-dependent enhancements in LDI ion yields of transition metals from aerosol particles. Using an optical parametric oscillator (OPO), we demonstrate that besides Fe, also the sparsely detected and biologically relevant trace metals Zn and Mn can be observed in anthropogenic particles with much higher sensitivity. We show that the resonant absorption of iron coincides with the spectrum of the field-deployable KrF-excimer laser and with the REMPI absorption spectra of most aromatic molecules. Thus, the enhanced detection sensitivity for metals can be combined with detailed spectra of aromatic substances via REMPI.. Finally, we demonstrate the application potential of the resonance effects in a field study comparing the KrF-excimer laser with a commonly used ArF-excimer laser for their Fe detection capabilities in ambient aerosols. We found that resonant LDI also reveals Fe-signatures in particle types that produced no Fe-signals upon non-resonant LDI, suggesting that the relevance of organic aerosols and salts as source for Fe might have been underestimated in earlier SPMS studies.

## 2 Methods

2.1 Single-particle mass spectrometer and optical setup

The basic SPMS-instrument (Hexin Instruments Ltd., Guangzhou, P.R. China and Photonion GmbH, Schwerin, Germany) is described in other publications (Li et al., 2011). Briefly, its instrumental layout is conceptually close to the ATOF-MS (Su et al., 2004) with an aerodynamic lens inlet and an optical sizing unit that comprises of a pair of 75 mW cw-lasers at a wavelength of 532 nm, ellipsoidal mirrors and photomultipliers. The dual-polarity mass spectrometer is designed in Z-TOF geometry, as introduced by (Pratt et al., 2009). For further details, e.g. the inlet particle transmission and detection efficiency, we refer to the literature (Li et al., 2011; Zhou et al., 2016). After the laboratory experiments, we implemented delayed ion extraction ($\Delta t=0.4$ µs) using high-voltage switches (HTS31-03-GSM, Behlke GmbH, Germany) to improve the peak quality in the ambient air experiments (Vera et al., 2005; Li et al., 2018). Major modifications to the commercial device

comprise the ionization laser and the optical setup. We equipped the instrument with both a tuneable laser system (optical parametric oscillator, OPO) and excimer lasers ($\lambda$=248 nm and $\lambda$=193 nm) and replaced the Nd:YAG solid-state laser ($\lambda$=266 nm, 4th harmonic frequency) that belongs to the instruments standard configuration. Apart from the wavelength, most beam parameters were comparable throughout the experiments, see Table 1 for details. The pulse energy was measured at the optical entrance and exit of the mass spectrometer and the position of the focal lens (f=200 mm) was adjusted to maintain a comparable spot area, respective intensity for all wavelength comparison experiments. The OPO wavelengths as well as the KrF-excimer laser spectrum were measured with a LRL-005 spectrometer (MK Photonics Inc. U.S.).

## 2.2 Data analysis

In the laboratory experiments, only particles with both a positive and negative ion spectrum, each showingat least two peaks above the noise level, were considered. Raw time-of-flight data was converted to mass spectra considering peak area within nominal mass resolution by custom software on Matlab platform (MathWorks Inc.). For particle classification in the ambient air study, we utilized the adaptive resonance theory neural network, ART-2a (Song et al., 1999) from the open-source toolkit FATES (Flexible Analysis Toolkit for the Exploration of SPMS data) (Sultana et al., 2017) with a learning rate 0.05, a vigilance factor of 0.8 and 20 iterations.

## 2.3 Model particles, sampling and setup for ambient air experiments

Diesel exhaust particles from an old van (Volkswagen Transporter 1.7 D, build 1988) were collected from the inner surface of the exhaust tube. These particles exhibit a rather uniform chemical composition as demonstrated in previous experiments (Passig et al., 2017; Schade et al., 2019). Model particles for mineral dust were Arizona test dust 03 µm diameter (Powder Technology Inc., U.S.) and complex anthropogenic aerosols with trace metals were mimicked using NIST urban dust 1649b (Gonzalez and Choquette, 2016). Using a turntable-based powder disperser (Model 3433, TSI Inc., U.S.), particles were introduced into a 1 l/min carrier gas stream ($N_2$, purity: 5.0) from which 0.1 l/min were guided in an isokinetic flow into the instrument. For the experiments on ambient air, the SPMS instrument was set up at a meteorological station in a rural environment at the Swedish west coast, about 30 km south of Gothenburg (coordinates 57°23'37.8"N, 11°54'51.4"E). Ambient air was sampled at a height of 7 m above ground (15 m above sea level). Aerosols from a 300 l/min intake airflow were concentrated into the 1 l/min carrier gas stream using a first virtual impactor device (Model 4240, MSP corp., U.S.). After passing a dryer (Model MD-700-12S-1, Perma Pure LLC, U.S.), they were further concentrated to 0.1 l/min in a second step directly at the SPMS aerodynamic lens inlet. The concentration is most effective for particles around 1 µm size, while it drops below 0.5 µm, see Supplemental Fig. S1 for a comparison of particle numbers in ambient air with and without using the concentrator. The two KrF and ArF excimer lasers used in this experiment were alternately triggered to particles using a custom electronic circuit based on a complex programmable logic device (Intel Max V) with 8.5 ns pin-to-pin delay and programmed using Very High Speed Integrated Circuit Hardware Description Language (VHDL). The excimer laser beams were focused from opposite sides onto the particle beam, see Table 1 and section 3.3 for details.

**Table 1. Light sources and details of the optical setup.**

| Laser source | Opolette HE 355 LD UV, Opotek LLC, U.S. | PhotonEx, Photonion GmbH, Germany | ATLEX-I 300, ATL GmbH, Germany |
|---|---|---|---|
| Laser medium | Optical Parametric Oscillator, Nd:YAG pumped | KrF gas (excimer) | ArF gas (excimer) |
| Wavelength (nm), photon energy (eV) | tuneable 210−2400 | 248, 4.99 | 193, 6.41 |
| Pulse duration (ns) | ≈5 | | |
| Beam size (mm) | Ø3 nearly Gaussian | 3x6 Gaussian x flat top | 3x6 Gaussian x flat top |
| Interaction spot distance to focus (mm) | ≈8−11 | 7 | 7 |
| Rayleigh length (mm) | ≈1.2−1.5 | 1.4 | 1.1 |
| Interaction spot size (µm) | Ø160 | 105×210 | 105×210 |
| Pulse energy (mJ) | 0.4 | 3 | |
| Pulse intensity at interaction spot (GW/cm$^2$) | 0.8 | 5 | |

## 3 Results and discussion

3.1 Resonance enhancements of Fe signals

We measured the Fe signals from diesel soot and Arizona desert dust particles as representative models of relevant anthropogenic and natural aerosols transporting Fe into the oceans. Figure 1 (a) and (b) show the respective mass spectra of positive and negative ions from LDI with two different wavelengths using the OPO. The mass spectra were accumulated over each 400 particles, without normalization or further processing. The observed peak broadening results mainly from accumulation over single particle spectra with varying ion energy and starting positions. Typical signatures for (diesel) engine emissions (Toner et al., 2006) are recognizable, e.g. clusters of elemental carbon (EC, from soot) and organic hydrocarbon fragments (OC) (Silva and Prather, 2000). Also, alkali metals are frequently detected due to their low ionization energy. The desert dust particles (Fig. 1b)) reveal typical mineral dust signals from metals and metal oxides (Sullivan et al., 2007; Dall'Osto et al., 2010). The slightly different laser wavelengths yield rather similar mass spectra. However, much stronger Fe-signals can be observed for 248.3 nm for both particle types (see insets in Fig. 1(a) and (b)). This wavelength matches the $3d^64s^2 \rightarrow 3d^64s4p$ transition of Fe atoms, a line that is also typically used for Fe determination in atomic absorption spectroscopy. As apparent from the histogram plots in Figure 1 (c) and (d), the enhancement effect is not resulting from some especial Fe-rich particles. Instead, most particles show higher Fe-signals at the resonance wavelength and the fraction of particles without Fe-signals drops considerably. However, the high Fe-content of Arizona dust particles (≈47%) often leads to saturated signals on the single-particle level. Even stronger saturation effects producing highly

corrupted Fe-peaks were observed for hematite, which is consequently not shown here. Because interferences with $CaO^+$ and organic fragments such as $C_3H_4O^+$ can affect the signal differences at m/z=56, the histograms show the signal of the $^{54}Fe$ isotope. Contributions from organic fragments to m/z=54 are assumed to be rather small, as apparent from the signal

strengths of principal fragments in the respective mass range at m/z=51, 53 and 55, see inset of Figure 1(a). However, such interferences might lead to a moderate underestimation of the resonance enhancement. A further resonance effect can be noticed for Lithium at the reference wavelength of 242.2 nm because of the $1s^2 2s \rightarrow 1s^2 7p$ transition close to this wavelength. Experimental results on hemoglobin powder, representing a particle model with uniform organic composition, are shown in Supplemental Fig. S4 and confirm the resonance enhancements for Fe.


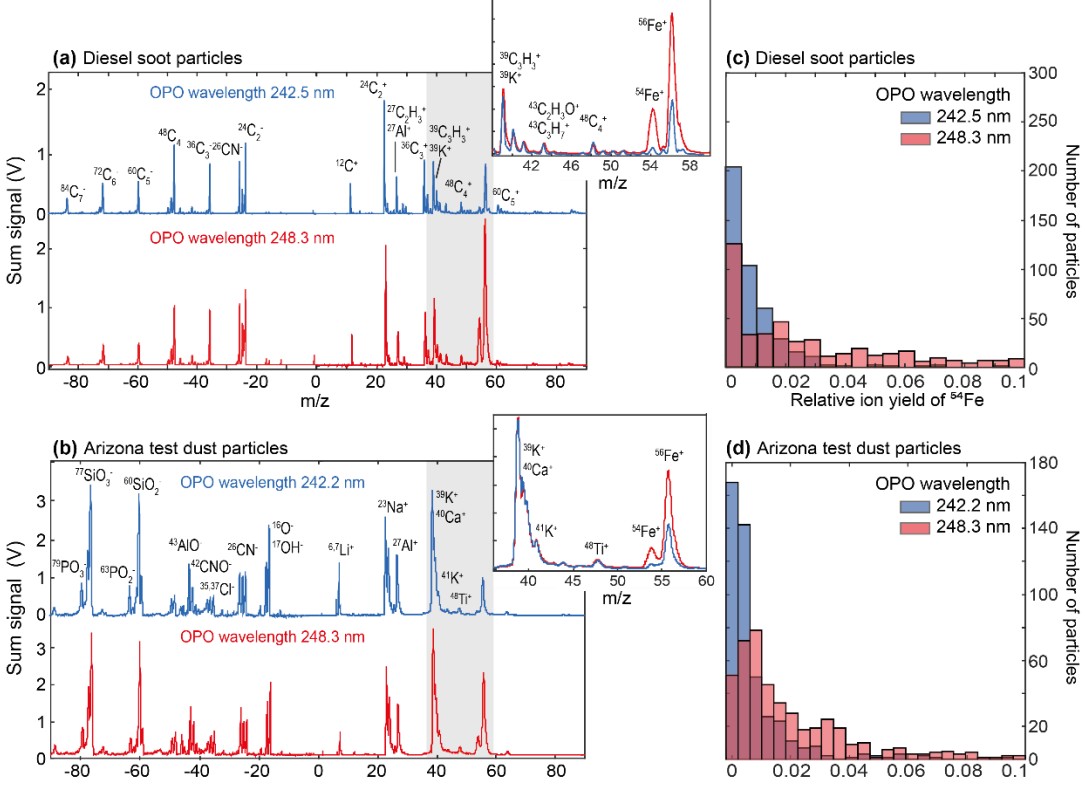

**Figure 1:** (a) Accumulated mass spectra (each n=400) of re-dispersed diesel soot particles ionized using the tuneable OPO laser. In the case of resonant ionization of Fe at 248.3 nm (red), the Fe signal is substantially enhanced compared to the non-resonant ionization at 242.5 nm (blue), see the inset for an enlarged view of the grey area. Most other signals are similar. (b) A comparable Fe-enhancement can

be observed for mineral dust particles. The histogram plots (c) and (d) of the single-particle relative ion signal ($^{54}Fe^+$ signal normalized to the particle's total ion yield) illustrate that the ionization enhancement accounts for the majority of analyzed particles. Corresponding normalized mass spectra are shown in Supplemental Fig. S2 and the particle size distributions are depicted in Supplemental Fig. S3.

To further investigate the enhancement effects, we measured the wavelength-dependent total ion yield of $^{54}Fe$ from each

1200 particles, exposed to OPO laser pulses of the same intensity. As shown in Figure 2, the maximum Fe-signal is achieved

near the resonance, with an enhancement of about 3-4 for diesel soot and mineral dust particles as well as for hemoglobin particles, see Supplemental Fig. S4. The ion yield curves have a remarkably width and are much broader than the atomic lines or the OPO-linewidth (4…6 cm$^{-1}$). The absorption spectrum of Fe-atoms (blue) represents data from the NIST atomic spectra library (Kramida and Ralchenko). Such signal enhancements at specific wavelengths were not reported in previous

SPMS studies, apart from the aforementioned REMPI-techniques. Thomson et al. (1997) observed for different salts, that the threshold intensity for ion formation decreased with increasing absorbance of the bulk material. Generally, more substances are ionized at higher photon energies and lower laser intensities are required, but these effects tend to saturate at higher laser intensities (Thomson et al., 1997; Murphy, 2007). Even in a study using two matrix-assisted LDI (MALDI) matrix materials absorbing at different wavelengths, Wade et al. (2008) found only minimal wavelength effects on ion yields but a stronger

dependence on the intensity and particle size. However, these results are not conflicting with the Fe resonance we observed. Several studies indicate that above a minimum intensity, effects in the plume dominate the ionization yield rather than the absorbance of the particle itself (Carson et al., 1997; Wade et al., 2008; Reinard and Johnston, 2008). The resonance begins to take effect as soon as Fe atoms are formed and vaporized from the particle during the initial phase of the laser pulse.

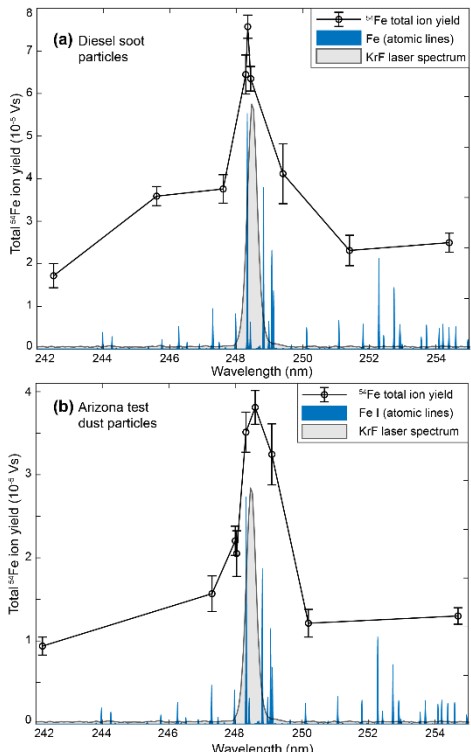

**Figure 2:** Wavelength-dependent total ion yield of $^{54}$Fe in SPMS of re-dispersed particles (black circles, n=1200, three replicates of 400 each). Both for (a) diesel soot particles and (b) Arizona desert dust particles, the signal peaks for wavelengths that match a major atomic transition of Fe (blue lines). The large width of the curve is attributed to line broadening through interaction with the dense particle surface. Coincidentally, the Fe-lines are also addressed by our KrF-excimer laser (measured spectrum in grey, arbitrary units). Atomic spectra from the NIST library (Kramida and Ralchenko). Mass spectra are shown in Figure 1, the respective curves of the normalized ion

signals and size distributions in Supplemental Fig. S3.

While so-far not recognized for SPMS, such resonance enhancements were previously reported and explained for laser ablation from solid surfaces. Using dye lasers, about five-fold signal increases were observed at the atomic lines of several metals and semiconductors (Verdun et al., 1987). The widths of the resonances were also rather broad, 0.4-0.7 nm. For low laser intensities, grazing incidence and two-step excitation, the width dropped below 0.05 nm (McLean et al., 1990) approaching the values of the respective atoms in gas phase ionization (Resonant Ionization MS, RIMS (Young et al., 1989)). The explanation for the broad signals in resonant ablation from surfaces and particles is rather simple: Broadening and transition wavelength shifts can be expected if the excitation happens when atoms are still bound in the matrix close to the surface (Verdun et al., 1987; McLean et al., 1990). Also the plasma pressure could contribute to these effects. With increasing time and distance from the dense target, the surface bonds vanish and the conditions become similar to RIMS. Minor contributions to the measured width could result from Stark broadening (typically at higher laser power (Hübert and Ankerhold, 2011)) and from interferences with the adjacent absorption lines.

### 3.2 Resonance enhancements of trace metals

The resonant ionization of particle-bound Fe raises the question whether the SPMS-based detection of other biologically relevant metals may also benefit from the enhancement. We used NIST Reference Material Urban Dust 1649b (National Standard Institute of Technology – U.S.) as a well-characterized anthropogenic particle model containing several transition metals at low concentrations. Figure 3 shows accumulated cation mass spectra from resonant and non-resonant ionization with respect to strong atomic lines of Fe, Mn and Zn. The mass fraction of Fe is rather high (≈3%) and the signal enhancement at 248.3 nm (see Fig. 3(a)) corresponds to the results from diesel soot and Arizona dust. Manganese contributes a mass fraction of only 0.024 % to the dust. In general, for particles with organic content, the Mn signature at m/z=55 can hardly be distinguished from molecular fragments of the same mass. However, when the OPO wavelength is in resonance with the $3d^54s^2 \rightarrow 3d^54s4p$ transition of Mn at 279.5 nm, a clear signal appears at m/z=55, nearly as high as the peak of the much more abundant $^{56}$Fe in the sum spectrum, see Fig. 3(b). Also for Zn (mass fraction 0.17%), there is a substantial difference and a clear signature appears in resonance case (Fig. 3(c)). Because the resonance wavelength of 213.8 nm is near the UV-limit of the OPO, the pulse energy of 0.25 mJ is lower than for the other metals and, in contrary to all other wavelength comparisons, the reference wavelength is higher than the resonance wavelength. After resonant excitation at the respective wavelength, the absorption of a further single photon is sufficient for ionization of all three metals. The histogram plots (d…f) prove that the enhancement does not only result from a minority of particles that contribute especially strong ion signals. In contrary, a higher number of individual particles reveal signatures of the respective metals, which indicates a more secure and sensitive detection. The results suggest that tuneable laser systems can be advantageous to enhance the detectability of various elements of interest in SPMS.

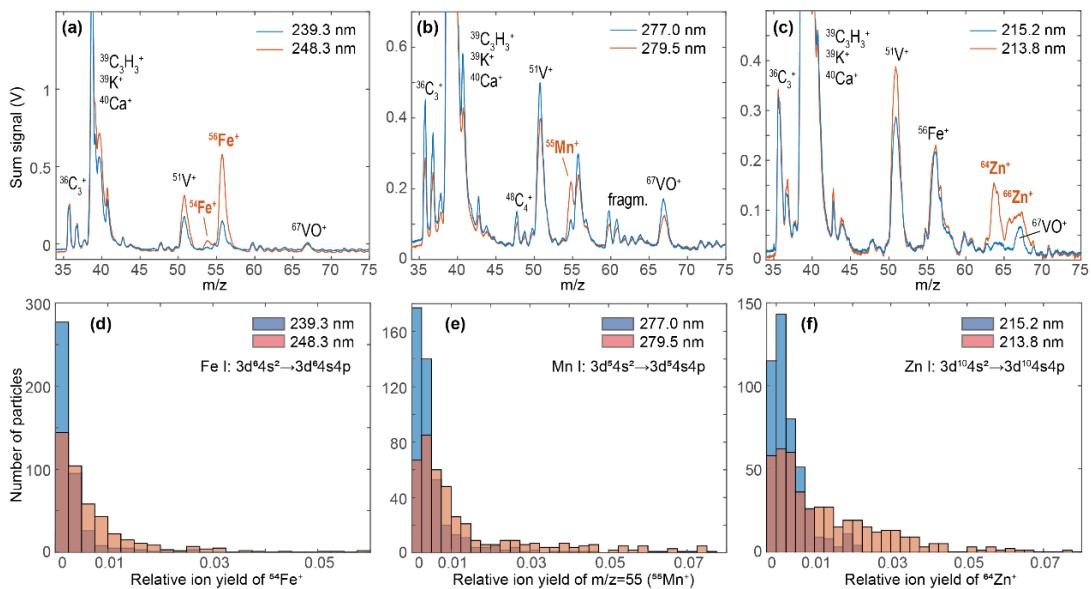

**Figure 3:** Accumulated cation mass spectra (n=400) of re-dispersed urban dust particles (Reference Material NIST 1649b). Using the tuneable OPO, the spectra were recorded at resonance wavelengths of each metal (red) and for the non-resonant case at a slightly different wavelength (blue). While carbon and molecular fragment signals are similar in the pairwise comparison, the resonant enhancements for (a) Fe, (b) Mn and (c) Zn are clearly visible. Complete, bipolar mass spectra and the size distribution are shown in Supplemental Fig. S5. (d-f) The single-particle distribution of the relative ion signals illustrates that the resonant ionization enhancement allows metal detection for many more particles. The respective resonance wavelengths (red) address the indicated transitions.

3.3 Application to long-range transported aerosols

While our laboratory experiments revealed remarkable resonance effects for several metals and particle types, these results have to be transferred into application on ambient aerosols. Tuneable laser systems are of limited suitability for field studies because of their complexity, low pulse power and repetition rate. In our experiments, thermal lensing problems of the irregularly triggered OPO system reduced its pulse power and stability, resulting in a shot-to-shot variability of the pulse power up to about 30 %. However, a freely triggerable OPO-SPMS with sufficient pulse energy is under development. In contrast to tuneable light sources, excimer lasers are cheaper, more robust and powerful. Of note, the KrF-excimer laser line at 248.3 nm coincidentally matches the strongest UV absorption line of Fe, a fact that gained little attention in the last decades (Trainor and Mani, 1978; Seder et al., 1986). The spectrum of our laser is shown in Fig. 2. We directly compare the Fe detection efficiencies of two field-deployable excimer lasers for the same ambient aerosol ensemble. The KrF-line is in resonance with the Fe absorption, while the often-used ArF-line is not. To exclude all effects from different instrumentation, both lasers are integrated into the same SPMS, firing with the same pulse parameters from opposite sites onto the particles, see Figure 4(a) and Table 1. A custom electronic circuit triggers the lasers alternately. With regard to the important application of detecting Fe-containing aerosols in remote regions, we designed our experiment to observe long-range transported anthropogenic particles with high secondary contributions in a marine environment. Therefore, we set up our

instrument at the Swedish west coast and measured aerosols from central Europe after transport over the Baltic Sea, see the back-trajectories in Fig. 4(b).

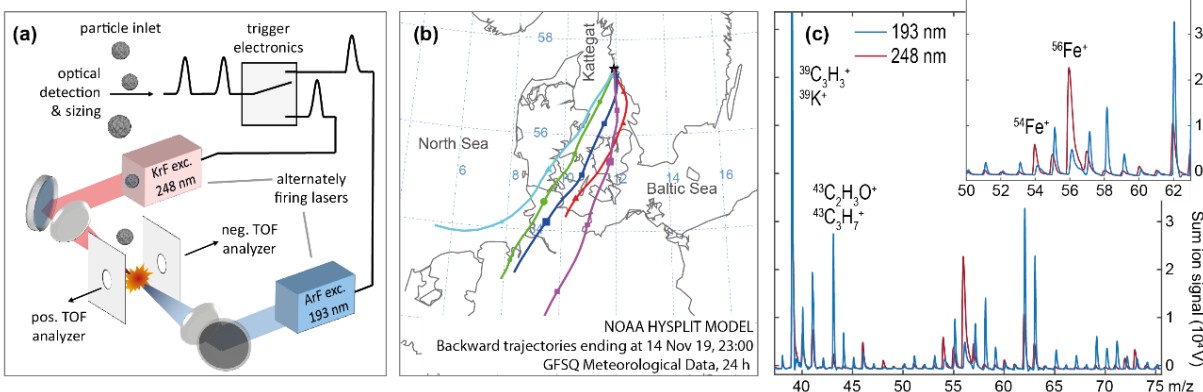

**Figure 4:** (a) Schematic view of the setup for direct comparison of non-resonant and resonant ionization of Fe in ambient air particles using the same mass spectrometer. The two lasers fired alternately on 15,000 particles each. (b) Back-trajectories from the HYSPLIT webtool (www.ready.noaa.gov/HYSPLIT.php), ending at the sampling site (sea level) during the experiment on long-range transported particles. (c) Accumulated cation mass spectra (each n=15,000) show a $Fe^+$-signal enhancement for ionization with the KrF excimer laser (248 nm, red) versus the ArF excimer laser (193 nm, blue). Further differences will be discussed in a different publication.

With each of the lasers, we analysed 15,000 individual particles on the 14 November 2019 between 15:00 and 24:00 local time. The mean particle mass concentration was 7.8 µg/m$^3$ (PM2.5) and 5.0 µg/m$^3$ (PM1.0) as measured by the station's dust monitor (Grimm EDM-180 MC). Figure 4(c) shows the resulting sum mass spectra of cations for each ionization wavelength. The enhanced Fe signature for the KrF-laser is clearly visible in the sum spectrum. All further wavelength-dependent differences will be discussed in a future publication. From each 15,000 particles exposed to the ArF-laser (KrF-laser), 13,776 (6,364) produced a negative spectrum, 12,217 (5,577) a positive signature and 12,189 (5,258) yielded bipolar mass spectra. The higher hit rate of the ArF laser results from the lower intensity thresholds for ion formation due to its higher photon energy (Thomson et al., 1997), thus yielding mass spectra also from particles that were not fully hit. Nearly all particles (>98 %) with negative spectra showed nitrate ($^{46}NO_2^-$ and $^{62}NO_3^-$). Because the steady onshore wind during the experiment excludes local sources of nitrate, these ions indicate condensation of $NO_3$ and replacement of $Cl^-$ by $NO_3^-$ (Gard et al., 1998; Arndt et al., 2017; Dall'Osto et al., 2016b) during long-range transport from central Europe (Dall'Osto et al., 2016a). Most single-particle spectra are dominated by either sea salt signatures ($^{23}Na^+$, $^{46}Na_2^+$, $^{62}Na_2O^+$, $^{63}Na_2OH^+$ and $^{81,83}Na_2Cl^+$) (Murphy et al., 2019), by organic fragments (e.g. $^{27}C_2H_3^+$, $^{39}C_3H_3^+$, $^{43}C_2H_3O^+$ and $^{43}C_3H_7^+$) (Silva and Prather, 2000) or they reveal internal mixtures of these main components. To investigate the Fe-enhancements on the single-particle level and to analyse the role of the particle's main components, we performed a cluster analysis for each set of bipolar single-particle spectra, excluding the mass channels m/z=54…56 that bear potential Fe-signatures. The ART-2a algorithm yielded 149 clusters for the particles ionized with the ArF-laser and 106 clusters for the KrF-laser ionization. Clusters with less than 20 particles were excluded from the analysis. Furthermore, clusters with comparable average mass spectra and the same major ions but slightly varying relative signal intensities were manually merged.

**Table 2. Main particle classes from ART-2a clustering and subsequent merging with respect to the major components. The respective mass spectra are shown in Supplemental Figures S6 and S7.**

| | Aged sea salt | Aged sea salt & minor OC | Salt/OC mixed | OC | OC+EC | Fe | Anions only |
|---|---|---|---|---|---|---|---|
| Dominating ion signals | $^{23}Na^+$, $^{46}Na_2^+$ $^{62}Na_2O^+$, $^{63}Na_2OH^+$, $^{46}NO_2^-$, $^{62}NO_3^-$ | | $^{23}Na^+$, $^{46}Na_2^+$, $^{39}K^+$ & mol. fragments | $^{39}K^+$, $^{43}C_2H_3O^+$ & molecul. fragments, $^{18}NH_4^+$, $^{30}NO^+$, $^{59}C_3H_9N^+$ (TMA) | | $^{56}Fe^+$, $^{73}FeOH^+$ | $^{46}NO_2^-$, $^{62}NO_3^-$ |
| Further required signals for assignment | $^{81,83}Na_2Cl^+$, $^{35,37}Cl^-$ | $^{39}K^+$ & molecul. fragments | balanced ratio between salt & OC signatures | no or minimal salt signatures | $^{24}C_2^+$, $^{36}C_3^+$, $^{24}C_2^-$, $^{36}C_3^-$, | | no cations |

The particle ensemble revealed six dominating particle groups, as summarized in Table 2. The corresponding ART-2a area matrices representing the average intensity for each m/z and thus reflecting the typical mass spectra within a group are shown in Supplement Figures S6 and S7. Further separation into subgroups, e.g. with respect to signals from $^{18}NH_4^+$, $^{30}NO^+$ or trimethylamine (TMA, m/z=58…59) (Healy et al., 2015; Köllner et al., 2017) had only limited effects on Fe detection and is consequently not shown here. Mineral dust particles were not observed in appreciable numbers. The measured size distribution is rather narrow, reflecting the instruments optimum detection efficiency that roughly coincides with the typical size mode undergoing long-range transport, see Supplemental Fig S8.

The particle numbers within the main classes are shown in Fig. 5. There are several differences between the two ionization wavelengths, e.g. the aforementioned overall hit rate. However, here we focus on the detection of Fe. In order to ensure a conservative effect registration (i.e. signals at m/z=56 may also stem from $CaO^+$ or molecular fragments such as $C_3H_4O^+$) Fe-content is only accounted for particles with a peak area at m/z=56 that is larger than both the signals at m/z=40 ($Ca^+$) and m/z=55 (principal fragment signal). To further strengthen the screening as recommended by previous studies (Zhang et al., 2014; Dall'Osto et al., 2016a), particles with an additional signal at m/z=54 from the $^{54}Fe$ isotope, which is lower than 1/10 of the peak area of $^{56}Fe$ are represented by black bars. Half of the particle spectra that were identified by the algorithm to show the $^{54}Fe$ isotope, were manually cross-checked on a random basis to prevent false positive results. From the 15,000 particles exposed to the 193 nm laser pulses, less than 100 particles show Fe signatures according to this stringent criterion. As apparent from the enlarged view on the right of Fig. 5(a), nearly all these particles revealed also strong carbon cluster signals from EC. This suggests that they either belong to a particular Fe-rich aerosol class, e.g. from ship emissions or that the EC matrix augments the ionization process of Fe (Zimmermann et al., 2003) in contrast to a salt/OC-matrix, where energetically preferred ions survive collisional charge transfer in the plume (Reinard and Johnston, 2008). Also a suppression of specific ions by the presence of water is conceivable (Neubauer et al., 1998), although a dryer was applied in our experiment. A very different Fe-detection was achieved with the resonant ionization at 248 nm, see Fig. 5(b). Even though the total particle hit rate was lower, many more particles with Fe-signatures were detected. A key finding is that the Fe-detection is not limited to particles with EC-signatures anymore, but the Fe appears to be internally mixed within particles of several classes. (The relatively low abundance of Fe in the OC-class can be explained by the high contribution of wood/biomass combustion particles.) Remarkably, many particles with low cation signals reveal nearly exclusively Fe-

signatures, providing an own group after further classification into subgroups (Fe-signatures were excluded from the first ART-2a clustering).

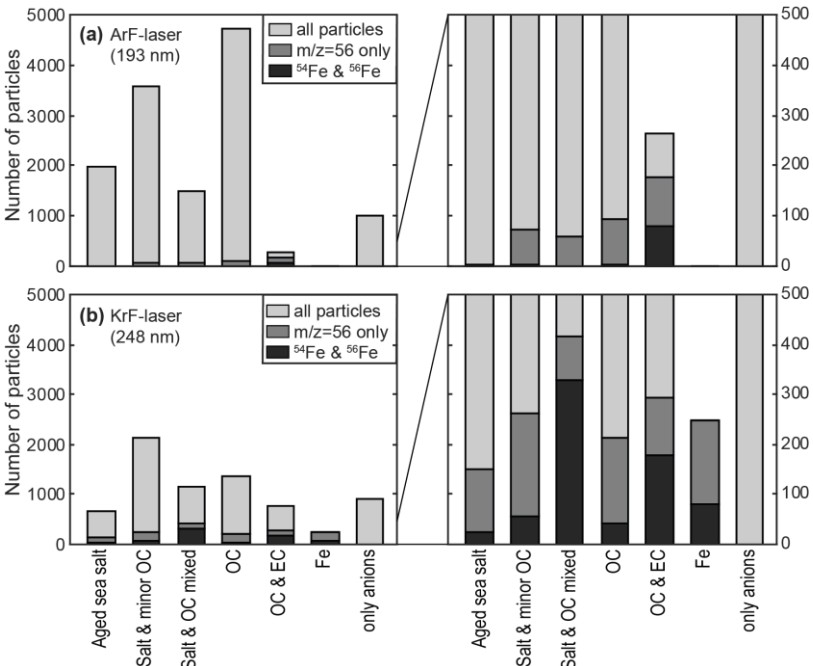

**Figure 5.** Number of particles within the main classes according to Table 2. Dark grey fractions represent particles with a peak area at
m/z=56 being larger than at m/z=55 (molecular fragments) and m/z=40 ($Ca^+$, because of interference with $^{56}CaO^+$), indicating Fe content. Black fractions illustrate particles showing an additional signal of the less abundant isotope $^{54}Fe^+$. (a) If ionized with 193 nm pulses, substantial fragmentation leads to dominating fragment signals in many of the 15,000 exposed particles. Fe-signals are almost exclusively observed for particles with EC-signatures (see the enlarged view on the right), indicating a particularly high Fe-content or possible interactions with strongly absorbing soot during ionization. (b) Although fewer particles produce ion signals if exposed to 248 nm pulses,
the particle fraction showing Fe-signatures is much larger and even a cluster with dominating Fe-signals appears. Of importance, the Fe-signals are not limited to EC-containing particles but can be observed for all classes. This suggests that the resonant ionization allows a more universal and secure detection of Fe.

Since the same aerosol ensemble was probed with both laser wavelengths, the appearance of Fe-signals for several particle matrices disagrees with the assumption of a particular Fe-rich class. In contrast, different ionization mechanisms are likely to
determine the Fe-detection, and resonant LDI appears to feature a more universal and secure detection approach for iron.

Although our field study provides only a limited dataset, some general implications can already be derived. The internal mixing of Fe with sulfate or organic acids is assumed to be crucial for Fe dissolution, and thus for the anthropogenic increase of bioavailable iron input to the oceans (Li et al., 2017). Previous studies indicated that the Fe transport into the sea is dominated by coal combustion particles containing sulfate in Asia (Furutani et al., 2011; Moffet et al., 2012), while the
majority of Fe-containing particles in Europe are mixed with nitrate and were attributed to traffic activities (Dall'Osto et al., 2016a). Similar to our experiment, these studies found strong internal mixing of many Fe-containing particles, such as biomass burning signals with coal combustion contributions in Asia and secondary nitrate with Fe in Europe. However, Fe-

particles with sea-salt signatures were negligible in the SPMS studies and mixtures of Fe and OC were a minor fraction (Furutani et al., 2011; Dall'Osto et al., 2016a). In our study, these particles were the most abundant types of Fe-containing

particles, if resonant ionization was applied, see Fig 5(b), while for non-resonant ionization, particles with EC signatures were dominant (Fig. 5(a)). Taking into account that the aforementioned SPMS studies utilized non-resonant LDI of Fe at 266 nm, Fe-transport in organic and salt/mixed aerosols might have been underestimated. Electron microscopy studies of individual particles in Asia frequently revealed thick coatings of secondary compounds and organic matter around Fe-rich particle components (Li et al., 2017; Moffet et al., 2012). Ultra-fine Fe-containing particles, such as soot from traffic

emissions, can enter the long-range transported accumulation mode via agglomeration with larger particles and condensation of organic vapours, secondary nitrate or sulfate. In our study, we observed a high prevalence of Fe in sea salt and OC particle types, indicating the importance of these pathways for transport of biologically relevant Fe.

**4 Conclusions**

In summary, we described enhancements in particle laser desorption/ionization that rely on resonant light absorption by

metal atoms. Combining laboratory and field experiments, we showed that the mechanism can be exploited to improve the detection of relevant metals in both natural and anthropogenic aerosols on the single-particle level. Not all physical details are fully understood, and the signal enhancement effects providing the basis for the improved metal detection efficiency are difficult to quantify for the different particle types. However, our results show that the increase in sensitivity is moderate for particular Fe-rich aerosols, such as the about 3-fold signal enhancement for Arizona test dust. The resonance enhancement

appears to become more effective for mixed particles with smaller Fe-contributions such as in the ambient air experiment, where about 10 times more particles revealed Fe-signatures in direct comparison with non-resonant ionization. Taking into account the lower hit rate of the KrF laser that is related to its lower photon energy, the overall efficiency to identify Fe signatures in a single-particle mass spectrum was increased by a factor of about 20 in our ambient air study.

The coincidental matching of the KrF laser line with a strong absorption of Fe atoms allows an easy and straightforward

application of the resonance effect in the field. For direct comparison of KrF with ArF lasers, it has to be considered that the lower photon energy of KrF laser is associated with a reduced hit rate and different mass spectral signatures of other particle components, e.g. organics. Further studies are required to evaluate these differences. Of note, because of its rather high pulse energy and the flat-top beam profile, the hit rate of the KrF laser was about 40−50 % in our experiment, which is still more than the values that are typically achieved with the most common laser line in SPMS, the Nd:YAG at 266 nm.

Exploiting the resonance effect for other metals than Fe requires a tuneable Nd:YAG-OPO system, which is, however, more difficult to operate.

With the improved detection of Fe and its inherent sensitivity to further key nutrients such as nitrate and phosphate, SPMS becomes an interesting complement to established methods for investigating atmospheric Fe transport. Moreover, several key parameters for the metal's bioavailability, including the particle size or the presence of carboxylic acids and sulfate

(Fang et al., 2017) can be determined on a single-particle level. Because of the high time resolution, SPMS-based Fe detection may be particularly helpful for studies on the oceans' rapid response to the naturally episodic depositions of Fe and other micronutrients. Beyond these direct applications, more studies are required to elucidate the promising implications for SPMS quantification approaches (Healy et al., 2013; Gemayel et al., 2017). Of note, the Fe-containing particles can further be characterized with regard to their organic content using multi-step ionization techniques (Schade et al., 2019; Czech et al.,

2017). This is of importance for health-related studies, as two of the most relevant adverse aerosol compounds, transition metals and PAHs, can be addressed with the same, easily accessible KrF-excimer laser wavelength.

Such hyphenated single-particle schemes bear great potential to elucidate intriguing interactions in atmospheric heterogeneous and multiphase chemistry (Pöschl and Shiraiwa, 2015), for example with regard to possible catalytic activities of the in particle's metal content (Sullivan et al., 2007). In conclusion, the described resonance effects pave a new route

towards improved detection of air pollutants and a more profound understanding of the aerosol impact on biogeochemical cycles and human health.

**Data availability**

Data are available on request from Johannes Passig (johannes.passig@uni-rostock.de).


**Author Contributions**

J.P. and J.S. contributed equally to this work. J.P. conceived the experiments. J.S., E.-I.R. J.P., T.K.-B. and R.I. performed the experiments. L.L., X.L. and Z.Z. provided the SPMS instrument as well as technical support. T.K.-B. developed the electronics. H.C., M.S., T.S. and R.Z. provided assistance with data interpretation. J.M. and H.F. hosted and supported the

field study. J.S., J.P. and E.-I.R. analyzed data and prepared the figures. J.P. wrote the manuscript with contributions from all authors.

**Competing interests**

The authors declare that they have no conflict of interest.

**Acknowledgements**

We thank Johan Mellqvist, John Conway, Lars Eriksson and co-workers from the Chalmers University of Technology and from the IVL Miljöinstitut for hosting the field experiments and their support.

The Project was funded by the German Research Foundation (ZI 764/6-1), by the German Federal Ministry for Economic Affairs and Energy (ZF4402101 ZG7), by the Helmholtz International Lab aeroHEALTH (www.aerohealth.eu) and by the Helmholtz Virtual Institute of Complex Molecular Systems in Environmental Health (www.hice-vi.eu).

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
