# Peer review of "Resonance-Enhanced Detection of Metals in Aerosols using Single Particle Mass Spectrometry"

_Atmospheric Chemistry and Physics, 2020_

## Referee Comment (RC1) · Anonymous Referee #1 · 21 Feb 2020

**Review of "Resonance-Enhanced Detection of Metals in Aerosols using Single Particle Mass Spectrometry"**

**General comments**

In this study, the authors combined laboratory experiments and field measurements to demonstrate resonant ionization enhancement of particle-bound metals, i.e., Fe, Mn, Zn, and Li, by single particle mass spectrometry (SPMS). The authors show a new way to improve the detection capabilities of single particle mass spectrometer to specific species in aerosol particles. A tuneable laser system was used in the laboratory to investigate the wavelength-dependent resonance effects, which rarely applied in SPMS. Given the resonant ionization with KrF excimer laser ($\lambda$=248 nm), ambient particle-bound iron can be detected with much higher sensitivity, such that its source information can be well preserved. Considering the profound impact of particle-bound metals in marine environments and for human health, this study is quite helpful also in a broader research field. However, the method description is inadequate, some laboratory measurements and more discussions are needed. The atmospheric implication section should be strengthened to better fit for this journal. Therefore, I recommend it to be published after major revisions.

**Major comments**

1. The authors state that resonant ionization of metals in single particles is becoming independent of the particle matrix and its atmospheric changes. This is an important point, however, there is not enough evidence to support it. Although the enhancement effect has been observed not only in Fe-rich but also in other Fe-containing particles, it doesn't mean that the particle matrix won't play a role. I would argue that the degree of enhancement may change in different particle types. This is actually shown in Fig 2: when comparing the wavelength at 248. 3 and ~247 nm, an enhancement of the Fe-54 signal is above 2 and below 2 for diesel soot and Arizona test dust, respectively. Please add more discussions, rephrase the abstract and the conclusion sections.

2. Experimental Section (consider to change the header to "Methods")

1) This study focused on both technical improvements and application, thus more description of the instrument should be added, even though it has been described in other publications. The following points should be addressed in the main manuscript and/or supporting information. Briefly describe the inlet system (inlet type? transmission efficiency of the inlet to standard samples, e.g., PSL?), particle detection region (what is the detection limit of particle size? Scattering efficiency?), LDI region (hit rate for ArF and KrF?), spectra generation (e.g., some descriptions of data analysis in section 3.3 should be moved to the method section), and what is the overall transmission efficiency/detection efficiency of this instrument to PSL particles or other standard samples? In the field measurement, a custom electronic circuit was applied to trigger the lasers alternately. What is the firing time for each one?

2) In the manuscript, "delayed ion extraction" was used and mentioned in section 3.3 Line 234, but not described in method section. Please add it at least in the supporting information.

3) Regarding surrogates. Why was the diesel soot chosen as the anthropogenic Fe-containing particles? I suggest that some standard samples with specific Fe signal, e.g., hematite, should also be measured. This allows to see

more prominent resonant enhancement and the ratio of Fe-56/Fe-54 can be determined, which could help to refine the enhancement effect for other particle types.

3. Results and discussion

1) The mass spectra are shown in an accumulated way, without normalization or further processing. However, the accumulation method is not able to reduce the shot-to-shot difference, which might bring big uncertainty to the results. As stated in the manuscript, the irregularly triggered OPO system reduced its pulse stability. In the ambient, more stable excimer lasers were used, but the ambient particles are more dispersed and complicated. Therefore, further processing is highly recommended. For individual particles, normalization to the total ion intensity can account for shot-to-shot variability (Hatch et al., 2014; Shen et al., 2018). Afterwards, averaging the normalized ones can give statistically more relevant results.

In order to further investigate the resonant enhancement, the authors measured the wavelength-dependent total ion yield of Fe-54 shown in Figure 2. It would be better to show the averaged relative ion yield with standard deviation, which may give an ion yield curve with less broadening.

2) Bias induced by organics and uncertainty of the results should be discussed. For diesel soot, m/z 56+ could also originate from organics e.g., $C_4H_8+$. Similar interference effects could rise from m/z 56 CaO+ for Arizona Test Dust. Therefore, the enhancement of Fe signals could be underestimated (based on the results that KrF excimer laser leads to ionization enhancement of Fe, but does not change or even reduce the signals of organics). It would be great if this can be discussed more. M/z 54+ is a more characteristic peak for iron, but it is rather weak compared to m/z 56+. Such weak signals can be close to the noise level. As mentioned above, measurement of some standard samples like hematite may provide useful information for discussion.

3) In section 3.1 and Fig. 1. Why were wavelengths at 250 nm and 242.2 nm chosen to compare with 248.3 nm for diesel soot and Arizona test dust, respectively? Please clarify the reasons (242.2 was chosen to show the enhancement effect for Li?). Please consider to show more wavelengths (e.g., <248, 248, >248) for comparison and keep consistency for different samples. Similar questions and comments apply to section 3.3 and Fig. 3.

4. Atmospheric implications and conclusions should be strengthened. Consider to add an implication section or implement it to the conclusion section as it is now. However, several points should be addressed. For the conclusions, the authors should answer e.g., how much enhancement (%) can be achieved? What about the uncertainty? What are the weaknesses of the technique? E.g., resonant ionization enhancement can only be obtained for a few metals simultaneously. In addition, the sensitivity to the other species, e.g., organics, might be reduced while it is enhanced to metals. Hence, tuneable laser systems are required. For the implications consider mentioning that particle-bound metals, such as iron, can also serve as catalyst or reactant in chemistry. Consider to add more discussions on how this study can help understand/investigate the role of particle-bound metal and/or metal oxides in atmospheric processes, e.g., heterogeneous chemistry, transportation, etc.

**Minor comments**

Page(P) 3 Line (L)64: Regarding typical ionization products, metals especially alkaline metals (e.g., Na, K) and mineral components such as silicate should be mentioned.

P3L65: Change "EC, OC" to "EC or OC".

P3L66: Why is it "unique" compared to the other kinds of source information?

P3L75: "Herein" would mislead the reader to think that the two-step technique was used in this study. However, one step LDI was used.

P4L101: What does "meaningful" mean? Please describe.

P4L106: "0-3 μm" change the short line to a long one "–" and mention the diameter type (physical?).

P4L107: "modelled" might cause ambiguity. Consider another word, e.g., mimic.

P4L112–115: Is the concentration factor (=3000?) stable? What was the upstream particle number concentration to the inlet of the instrument? What were the particle number and mass concentrations during the measurement time?

P5L120: Table1 Change "x" to multiplication sign "×". Change "…" to "to" or "–" and throughout the manuscript.

P5L127: Aren't those accumulated spectra rather than averaged ones? I am confused here.

P6L132: "much stronger Fe-signals". Please quantify such enhancement. How much % stronger?

P6L141: In the histogram plots of Fig.1, what is the difference between the light red and dark red? Please add descriptions in the figure caption as well. Similar comment is applied to Fig. 3.

P6L147: This is the first time to mention supporting information, thus it should be Fig. S1 rather than S4. Please revise accordingly throughout the manuscript.

P7L151: Such signal enhancement of 4 is compared to which value at which wavelength? As shown in Fig2, when compared to the lowest value, they are 7 for diesel soot and 4 for mineral dust; comparing the wavelength at 248. 3 and ~247 nm, an enhancement of Fe-54 signal is >2 and <2 for diesel soot and Arizona test dust, respectively.

P7L155–160: Change this sentence to "Thomson et al. (1997) observed ….bulk material." Similarly, please revise the sentence "Wade et al…".

P7L158: Give the full name of "MALDI".

P7L160: Add reference.

P7L164: Fig. 2    please use the same scale for OPO wavelength in panel (a) and (b). Add Y axis with the corresponding units for atomic absorption spectrum and for KrF laser spectrum.

P7L170–172: Move to introduction.

P7L178: Give the full name of "RIMS".

P7L183–187: Move to introduction.

P8L192–193: Please keep consistency, either use % or mg/kg for mass fraction.

P10L244: If refer to organics, m/z 39+ should be $C_3H_3^+$.

P10L245–252: Move the data analysis part to the method section.

P11L259: Add more references for TMA, e.g., Angelino et al., 2001; Pratt et al., 2009; Köllner et al., 2017.

P11L269–270: What it the threshold of Fe signal(s) to identify Fe-containing particles.

P12L290–292: Fe can be detected in different particle types does not suggest less particle matrix effect. Please see the first major comment and rephrase the sentence, as well as the following statement.

Supplement Figure S4: Can you really detect particles in the size range from 0 to 200 nm? What is the detection limit of this instrument? Please add such information in the method section in the main manuscript, and only show the data within the detection limits.

**References:**

Angelino, S., Suess, D. T., and Prather, K. A.: Formation of aerosol particles from reactions of secondary and tertiary alkylamines: Characterization by aerosol time-of-flight mass spectrometry, Environ Sci Technol, 35, 3130–3138, 2001.

Hatch, L. E., Pratt, K. A., Huffman, J. A., Jimenez, J. L., and Prather, K. A.: Impacts of aerosol aging on laser desorption/ ionization in single-particle mass spectrometers, Aerosol Sci. Tech., 48, 1050–1058, 2014.

Köllner, F., Schneider, J., Willis, M. D., Klimach, T., Helleis, F., Bozem, H., Kunkel, D., Hoor, P., Burkart, J., Leaitch, W. R., Aliabadi, A. A., Abbatt, J. P. D., Herber, A. B., and Borrmann, S.: Particulate trimethylamine in the summertime Canadian high Arctic lower troposphere, Atmos Chem Phys, 17, 13747–13766, 2017.

Pratt, K. A., Hatch, L. E., and Prather, K. A.: Seasonal Volatility Dependence of Ambient Particle Phase Amines, Environ Sci Technol, 43, 5276–5281, 2009.

Shen, X., Ramisetty, R., Mohr, C., Huang, W., Leisner, T., and Saathoff, H.: Laser ablation aerosol particle time-of-flight mass spectrometer (LAAPTOF): Performance, reference spectra and classification of atmospheric samples. Atmos. Meas. Tech., 11, 2325–2343, 2018.

---

## Author Comment (AC1) · 17 Mar 2020

We thank the referee for the report and the valuable comments.

With the Covid-19 outbreak in Europe, our labs were closed and currently we cannot conduct any experiments and also cannot access our experimental data. We apologize for the delay and will continue to work on the revised manuscript as soon as possible.

Thank you for your patience!

---

## Referee Comment (RC2) · Anonymous Referee #2 · 18 Apr 2020

The authors present a manuscript that describes the investigation of enhanced resonance ionisation as a means of improving the detection of metals in aerosol particles with single-particle mass spectrometry (SPMS). This is a very worthwhile objective, as such a technique could improve the online study of the distribution and sources of important metal species in the atmosphere. The authors focus on the resonance enhancement of Iron using laser desorption ionisation (LDI) at 248nm, which coincides with a major absorption line of Fe, which is an element of relevance in the study of micronutrients to the oceans and anthropogenic aerosol pollution. The physical basis for the techniques is sound and the work represents a substantial investment in resources to achieve a demonstrable improvement in detection of Fe along with Zn and Mn.

[Figure]

With the acceptation of some minor improvements in the reporting of the results, the technical merits of the manuscript is good. However there are some major concerns about the atmospheric relevance of the field study presented. Whilst this field data supports the technical development, it does not offer insight into the atmospheric implications of the presence of Fe in the environment in which it was measured. As this is a key requirement of this journal, the authors are requested to provide this discussion or consider submitting to a technical journal such as AMT.

Please also note the supplement to this comment:
https://www.atmos-chem-phys-discuss.net/acp-2020-25/acp-2020-25-RC2-supplement.pdf

[Figure]

**Supplement:**

The authors present a manuscript that describes the investigation of enhanced resonance ionisation as a means of improving the detection of metals in aerosol particles with single-particle mass spectrometry (SPMS). This is a very worthwhile objective, as such a technique could improve the online study of the distribution and sources of important metal species in the atmosphere. The authors focus on the resonance enhancement of Iron using laser desorption ionisation (LDI) at 248nm, which coincides with a major absorption line of Fe, which is an element of relevance in the study of micronutrients to the oceans and anthropogenic aerosol pollution. The physical basis for the techniques is sound and the work represents a substantial investment in resources to achieve a demonstrable improvement in detection of Fe along with Zn and Mn.

With the acceptation of some minor improvements in the reporting of the results, the technical merits of the manuscript is good. However there are some major concerns about the atmospheric relevance of the field study presented. Whilst this field data supports the technical development, it does not offer insight into the atmospheric implications of the presence of Fe in the environment in which it was measured. As this is a key requirement of this journal, the authors are requested to provide this discussion or consider submitting to a technical journal such as AMT.

Minor Comments:

L51 Most of the iron in mineral dust is in the form Fe2O3 (hematite) which is also insoluble. A distinction between soluble and insoluble Fe should be made as it affects bioavailiblity which is a major justification for carrying out this work.

L59    Replace 'were' with 'have been'.

L61    Replace/remove 'herein', as it implies it is a feature if this study.

L63    Not all SPMS is bipolar, e.g. PALMS.

L64    Replace 'typical ionization products' with 'typical observed ions'

L65    It would be useful to be more specific about which metals have been detected. Provide some references.

L68    The matrix effect will influence all compounds, not just the minor ones. The description of the matrix effect should be developed further.

L75    The description of LDI and REMPI are a little conflated. It would be easier to follow if they were described in separate paragraphs. It should also be clear when talking about the techniques generally, or specifically to SPMS. The reference Gunzer *et al* (2019) is a review paper not specific to SPMS for example.

L87    The conclusions of the present study and those of Schade et al (2019) should be clearly separated.

**Experimental Section**    An introductory paragraph at the start of this section would be useful to set the scene for the experimental approached that follow. A better description of the instrument geometry should be provided. The authors reference an early technical paper for the Hexin instrument (Li *et al* 2011) but it is unclear what other modification have been made (if any) that could influence the instrument performance. Are any prior publication made with this instrument platform (e.g. Schade et al 2019)?

**Results and Discussion**  A short introductory paragraph at the start of this section would also be helpful. The subsection titles should have some equivalence e.g. 3.2 Resonance enhancement of trace metals.

In the lab experiments, why were different OPO wavelengths used for the soot (250.0nm) and test dust (242.2nm)?

Figure1 What is the right hand axis of panel (B)?

L152 'remarkably width'

L182-186 Seems to be introductory material. Could this be moved to the introduction?

L190     The NIST reference material is described as 'well characterised'. A summary of this characterisation should be given e.g. from NIST certificate of analysis https://www-s.nist.gov/srmors/certificates/1649b.pdf

L200     'prove that not only the sum signals of the metals are higher in resonant case, but also more individual particles reveal their signatures.' This sentence is unclear and should be re-written with reference to limit of detection.

L213     Data that describes this instability should be provided.

Figure 4 Caption          'The two lasers fired alternately on 1500 particle each.'

**Conclusion** Some quantitative estimation of the enhancement achieved should be given. The trade-off between metal enhancement and lower particle detection rates should be highlighted.

**References**

Gunzer, F., Krüger, S., and Grotemeyer, J.: Photoionization and photofragmentation in mass spectrometry with visible and UV lasers, Mass Spectrom. Rev., 38, 202–217, doi:10.1002/mas.21579, 2019.

Schade, J., Passig, J., Irsig, R., Ehlert, S., Sklorz, M., Adam, T., Li, C., Rudich, Y., and Zimmermann, R.: Spatially Shaped 475 Laser Pulses for the Simultaneous Detection of Polycyclic Aromatic Hydrocarbons as well as Positive and Negative Inorganic Ions in Single Particle Mass Spectrometry, Anal. Chem., 91, 10282–10288, doi:10.1021/acs.analchem.9b02477, 2019.

Li, L., Huang, Z., Dong, J., Li, M., Gao, W., Nian, H., Fu, Z., Zhang, G., Bi, X., Cheng, P., and Zhou, Z.: Real time bipolar time-of-flight mass spectrometer for analyzing single aerosol particles, Int. J. Mass Spectrom., 303, 118–124, doi:10.1016/j.ijms.2011.01.017, 2011.

---

## Author Response (AR1)

Response to Anonymous Reviewer #1
We thank the reviewer for his work and the valuable comments. We are convinced that addressing the issues raised by the reviewer considerably improved the manuscript. Please see our reply below.

Note:
*Reviewer comments are in italics.*
Author responses are in normal format.
**Changes** that were made to the manuscript are in **bold** face with examples of the new text in blue.

*1. The authors state that resonant ionization of metals in single particles is becoming independent of the particle matrix and its atmospheric changes. This is an important point, however, there is not enough evidence to support it. Although the enhancement effect has been observed not only in Fe-rich but also in other Fe-containing particles, it doesn't mean that the particle matrix won't play a role. I would argue that the degree of enhancement may change in different particle types. This is actually shown in Fig 2: when comparing the wavelength at 248. 3 and ~247 nm, an enhancement of the Fe-54 signal is above 2 and below 2 for diesel soot and Arizona test dust, respectively. Please add more discussions, rephrase the abstract and the conclusion sections.*

We agree that the matrix still plays a role, especially during heating and desorption within the leading edge of the laser pulse and before the resonance with free atoms becomes effective. In our ambient air experiment, where the same particle ensemble was probed with two different wavelengths, we see that Fe-detection was restricted to a particular matrix (EC) in non-resonant LDI, which is in line with other studies that the strongly absorbing soot components augment ionization (e.g. Zimmermann et al., 2003). For resonant LDI, also particles without EC signature, but OC and even strong sea salt contribution revealed Fe. Reinard et al. (2008) showed that energetically preferred ions survive and thus dominate the spectrum, which explains the substantial fraction of the matrix effect that results from charge transfer in the plume. This is in line with our explanation, that strong light absorption enhances the degree of ionization for Fe which can therefore be detected even if high abundances of e.g. Na+ and K+ ions are in the plume.

**changes made to the manuscript:**
**Following the reviewer's comment, we rephrased the discussion on matrix effects throughout the manuscript and avoid the term "matrix effect" for our discussion.**
- **abstract: statement on dependency on the matrix effect removed. New sentence:**
  "Many of the particles that showed iron contents upon resonant LDI were mixtures of sea salt and organic carbon. For non-resonant ionization, iron was exclusively detected in particles with a soot contribution. This suggests that resonant LDI allows a more universal and secure metal detection in SPMS."
- **caption Fig. 5: statement on matrix effect dependency removed.**
- **Discussion of ambient air experiment. Sentence on matrix effects as well as comparison to REMPI and quantification removed.**

*2. Experimental Section (consider to change the header to "Methods")*
*1) This study focused on both technical improvements and application, thus more description of the instrument should be added, even though it has been described in other publications. The following points should be addressed in the main manuscript and/or supporting information. Briefly describe the inlet system (inlet type? transmission efficiency of the inlet to standard samples, e.g., PSL?), particle detection region (what is the detection limit of particle size? Scattering efficiency?), LDI region (hit rate for ArF and KrF?), spectra generation (e.g., some descriptions of data analysis in section 3.3 should be moved to the method section), and what is the overall transmission efficiency/detection efficiency of this instrument to PSL particles or other standard samples? In the field measurement, a custom electronic circuit was applied to trigger the lasers alternately. What is the firing time for each one?*

**changes made to the manuscript:**
**We added the requested technical information, further references and re-structured the "Methods" section accordingly.**
- **example on new text on the basic instrumental design**
  "Briefly, its instrumental layout is conceptually close to the ATOF-MS (Su et al., 2004) with an aerodynamic lens inlet and an optical sizing unit that comprises of a pair of 75 mW cw-lasers at a wavelength of 532 nm, ellipsoidal mirrors and photomultipliers. The dual-polarity mass spectrometer is

designed in Z-TOF geometry, as introduced by (Pratt et al., 2009). For further details, e.g. the inlet particle transmission and detection efficiency, we refer to the literature (Li et al., 2011; Zhou et al., 2016). After the laboratory experiments, we implemented delayed ion extraction ($\Delta t=0.4$ μs) using high-voltage switches (HTS31-03-GSM, Behlke GmbH, Germany) to improve the peak quality in the ambient air experiments (Vera et al., 2005; Li et al., 2018)"

- The hit rates for the ArF and KrF lasers are shown in Fig. 5.
- **on the electronics and firing time:** "The two KrF and ArF excimer lasers used in this experiment were alternately triggered to particles using a custom electronic circuit based on a complex programmable logic device (Intel Max V) with 8.5 ns pin-to-pin delay and programmed using Very High Speed Integrated Circuit Hardware Description Language (VHDL)."

*2) In the manuscript, "delayed ion extraction" was used and mentioned in section 3.3 Line 234, but not described in method section. Please add it at least in the supporting information.*

**Information added. See previous item.**

*3) Regarding surrogates. Why was the diesel soot chosen as the anthropogenic Fe-containing particles? I suggest that some standard samples with specific Fe signal, e.g., hematite, should also be measured. This allows to see 2 more prominent resonant enhancement and the ratio of Fe-56/Fe-54 can be determined, which could help to refine the enhancement effect for other particle types.*

In our study, we focused on the practical implications of the resonance for ambient air applications, and thus we tried to choose realistic proxies for the most relevant aerosols. We also worked with different particle types such as hematite. Unfortunately, the Fe signal was too strong in Fe-rich aerosols such as hematite, which lead to massive saturation effects, as already mentioned for Arizona test dust. Even with reduced detector voltages, the dynamic range of our SPMS was not sufficient to achieve a good peak quality, while preserving neighboured signals of other ions. Following the advice of the referee, we tried further particle systems and had some success with hemoglobin powder. We added this results in the Supplement. There were no general differences to the diesel or dust particles and the resonance effect was also observed for hemoglobin.

**changes made to the manuscript:**
**We performed new experiments with hematite and hemoglobin, added the results on hemoglobin in the Supplemental Fig. S4 and refer to it in the manuscript.**

3. Results and discussion
*1) The mass spectra are shown in an accumulated way, without normalization or further processing. However, the accumulation method is not able to reduce the shot-to-shot difference, which might bring big uncertainty to the results. As stated in the manuscript, the irregularly triggered OPO system reduced its pulse stability. In the ambient, more stable excimer lasers were used, but the ambient particles are more dispersed and complicated. Therefore, further processing is highly recommended. For individual particles, normalization to the total ion intensity can account for shot-to-shot variability (Hatch et al., 2014; Shen et al., 2018). Afterwards, averaging the normalized ones can give statistically more relevant results.*

With the accumulated, unprocessed spectra, we intend to show the magnitude of the physical resonance effect, i.e. the increase of ion current. This information would be biased by normalization. We agree with the reviewer, that the increase in detection efficiency can better be depicted by the signal normalized to total ion signal of each particle. This is exactly what we did in the histograms, where the relative signal of $^{54}$Fe normalized to total ion signal is shown, revealing the single-particle distribution of this signal and thus giving an impression of the detection frequency for Fe. We would prefer to avoid large additional mass spectra for normalized data in the manuscript and prepared those plots for the Supplement.

**changes made to the manuscript:**
- **We added normalized spectra for all model particles in the Supplement (Figs. S2-S5) and referred to in the manuscript.**

*In order to further investigate the resonant enhancement, the authors measured the wavelength-dependent total ion yield of Fe-54 shown in Figure 2. It would be better to show the averaged relative ion yield with standard deviation, which may give an ion yield curve with less broadening.*

Following the advice of the reviewer, we performed the experiment again and made 3 replicates with each 400 analysed particles for every wavelength. This allowed us to add standard deviation error bars.

Regarding the normalization, the argumentation is the same as for the previous issue. To account for the reviewer's comments, we prepared the same plots with the signal normalized to total ion counts and added it in the Supplemental.

**changes made to the manuscript:**
- **new data measured for diesel soot and Arizona dust**
- **numbers of analysed particles per wavelength increased from 400 to 1200**
- **error bars included**
- **new Figure 2**
- **new Figure S3 with normalized data.**

*2) Bias induced by organics and uncertainty of the results should be discussed. For diesel soot, m/z 56+ could also originate from organics e.g., C4H8+. Similar interference effects could rise from m/z 56 CaO+ for Arizona Test Dust. Therefore, the enhancement of Fe signals could be underestimated (based on the results that KrF excimer laser leads to ionization enhancement of Fe, but does not change or even reduce the signals of organics). It would be great if this can be discussed more. M/z 54+ is a more characteristic peak for iron, but it is rather weak compared to m/z 56+. Such weak signals can be close to the noise level. As mentioned above, measurement of some standard samples like hematite may provide useful information for discussion.*

According to the reviewer's advice, we extended the discussions on such interferences. We agree that these interferences might lead to an underestimation of the resonance effect, but it appears impossible to avoid it for realistic aerosols without substantially higher mass resolution. We prefer to leave the assessment of the effect strengths to the reader by providing the data in the figures, as this is quite difficult and differs across the particle types.
As mentioned before, we did additional experiments with other particles and added the hemoglobin results in the Supplement.

**changes made to the manuscript:**
- **Discussion on interferences and possible underestimation of the enhancement effect added and emphasised throughout the manuscript.**
  "Even stronger saturation effects producing highly corrupted Fe-peaks were observed for hematite, which is consequently not shown here. Because interferences with $CaO^+$ and organic fragments like $C_3H_4O^+$ can affect the signal differences at m/z=56, the histograms show the signal of the $^{54}$Fe isotope. Contributions from organic fragments to m/z=54 are assumed to be rather small, as apparent from the signal strengths of principal fragments in the respective mass range at m/z=51, 53 and 55, see inset of Figure 1(a). However, such interferences might lead to a moderate underestimation of the resonance enhancement."
- **new data measured for hemoglobin and presented in Fig. S4 as mentioned before.**

*3) In section 3.1 and Fig. 1. Why were wavelengths at 250 nm and 242.2 nm chosen to compare with 248.3 nm for diesel soot and Arizona test dust, respectively? Please clarify the reasons (242.2 was chosen to show the enhancement effect for Li?). Please consider to show more wavelengths (e.g., <248, 248, >248) for comparison and keep consistency for different samples. Similar questions and comments apply to section 3.3 and Fig. 3.*

We thank the reviewer for his attentive look. Indeed, a reference wavelength at 242 nm was foreseen, the 250 nm reference is our mistake. We corrected the figure and text accordingly, that both for the soot and the dust the same wavelength are compared (apart from a small deviation of 0.3 nm related to the OPO/spectrometer inaccuracy). Now, in all spectra, the reference wavelength is shorter, i.e. the photon energy is higher. Thus, possible effects of a higher photon energy, e.g. photoionization thresholds, are excluded and the enhancement can be attributed to the resonance with a higher level of certainty. Only for Zn, the reference wavelength is longer, which is associated with the instrument's UV limit and explained in the text.
Moreover, we added new data, comparing the Fe, Mn and Zn yield between three wavelengths (below resonance, at res. and above res.), as suggested by the reviewer. Because such plots showing three spectra are quite complex, we added them in the Supplement. For soot and Arizona dust, an extensive wavelength comparison is shown in Figure 2.

**changes made to the manuscript:**
- **new Figure 1 now shows a consistent wavelength comparison for Diesel soot and Arizona dust.**
- **statement on the exception of a higher wavelength for Zn added in the text.**
- **new data shown in Supplemental Figure S5 presents three-wavelength comparison for all Fe, Mn and Zn in NIST urban dust.**

*4. Atmospheric implications and conclusions should be strengthened. Consider to add an implication section or implement it to the conclusion section as it is now. However, several points should be addressed. For the conclusions, the authors should answer e.g., how much enhancement (%) can be achieved? What about the uncertainty? What are the weaknesses of the technique? E.g., resonant ionization enhancement can only be obtained for a few metals simultaneously. In addition, the sensitivity to the other species, e.g., organics, might be reduced while it is enhanced to metals. Hence, tuneable laser systems are required. For the implications consider mentioning that particle-bound metals, such as iron, can also serve as catalyst or reactant in chemistry. Consider to add more discussions on how this study can help understand/investigate the role of particle-bound metal and/or metal oxides in atmospheric processes, e.g., heterogeneous chemistry, transportation, etc. 3*

We thank the reviewer for considering our results generally important and the suggestions. We added a new discussion part to the ambient air results, focusing on the Fe-content of salt and organic aerosols we observed. We discuss these results in the context of previous SPMS studies. Moreover we extended the discussion and conclusions on atmospheric implications as suggested by the reviewer.

**changes made to the manuscript:**

- **new part discussing the mixing state of Fe-containing particles and implications for future studies. Additional text and references.**

"Although our field study provides only a limited dataset, some general implications can already be derived. The internal mixing of Fe with sulfate or organic acids is assumed to be crucial for Fe dissolution, and thus for the anthropogenic increase of bioavailable iron input to the oceans (Li et al., 2017). Previous studies indicated that the Fe transport into the sea is dominated by coal combustion particles containing sulfate in Asia (Furutani et al., 2011; Moffet et al., 2012), while the majority of Fe-containing particles in Europe are mixed with nitrate and were attributed to traffic activities (Dall'Osto et al., 2016). Similar to our experiment, these studies found strong internal mixing of many Fe-containing particles, such as biomass burning signals with coal combustion contributions in Asia and secondary nitrate with Fe in Europe. However, Fe-particles with sea-salt signatures were negligible in the SPMS studies and mixtures of Fe and OC were a minor fraction (Furutani et al., 2011; Dall'Osto et al., 2016). In our study, these particles were the most abundant types of Fe-containing particles, if resonant ionization was applied, see Fig 5(b), while for non-resonant ionization, particles with EC signatures were dominant (Fig. 5(a)). Taking into account that the aforementioned SPMS studies utilized non-resonant LDI of Fe at 266 nm, Fe-transport in organic and salt/mixed aerosols might have been underestimated. Electron microscopy studies of individual particles in Asia frequently revealed thick coatings of secondary compounds and organic matter around Fe-rich particle components (Li et al., 2017; Moffet et al., 2012). Ultra-fine Fe-containing particles, such as soot from traffic emissions, can enter the long-range transported accumulation mode via agglomeration with larger particles and condensation of organic vapours, secondary nitrate or sulfate. In our study, we observed a high prevalence of Fe in sea salt and OC particle types, indicating the importance of these pathways for transport of biologically relevant Fe."

- **Conclusion section extended to better address the general implications for atmospheric chemistry.**

"Not all physical details are fully understood, and the signal enhancement effects providing the basis for the improved metal detection efficiency are difficult to quantify for the different particle types. However, our results show that the increase in sensitivity is moderate for particular Fe-rich aerosols, such as the about 3-fold signal enhancement for Arizona test dust. The resonance enhancement appears to become more effective for mixed particles with smaller Fe-contributions such as in the ambient air experiment, where about 10 times more particles revealed Fe-signatures in direct comparison with non-resonant ionization. Taking into account the lower hit rate of the KrF laser, a downside related to its lower photon energy, the overall efficiency to identify Fe signatures in a single-particle mass spectrum was increased by a factor of about 20 in our ambient air study.

The coincidental matching of the KrF laser line with a strong absorption of Fe atoms allows an easy and straightforward application of the resonance effect in the field. For direct comparison of KrF with ArF lasers, it has to be considered that the lower photon energy of KrF laser is associated with a reduced hit rate and different mass spectral signatures of other particle components, e.g. organics. Further studies are required to evaluate these differences. Of note, because of its rather high pulse energy and the flat-top beam profile, the hit rate of the KrF laser was about 40−50 % in our experiment, which is still more than the values that are typically achieved with the most common laser line in SPMS, the Nd:YAG at 266 nm. Exploiting the resonance effect for other metals than Fe requires a tuneable Nd:YAG-OPO system, which is, however, more difficult to operate."

**Minor comments**

*Page(P) 3 Line (L)64: Regarding typical ionization products, metals especially alkaline metals (e.g., Na, K) and mineral components such as silicate should be mentioned.*

**changed accordingly**

*P3L65: Change "EC, OC" to "EC or OC".*

**corrected**

*P3L66: Why is it "unique" compared to the other kinds of source information?*

**"unique" should emphasise the combination of single-particle information and metal detection. Changed "Beyond" to "Along with". Added examples for source apportionment using metal signatures.**

*P3L75: "Herein" would mislead the reader to think that the two-step technique was used in this study. However, one step LDI was used.*

**corrected**

*P4L101: What does "meaningful" mean? Please describe.*

**"Meaningful" removed, conditions described.**

*P4L106: "0-3 μm" change the short line to a long one "−" and mention the diameter type (physical?).*

**This is the official name of the dust, we added "diameter**", since we could not find the information to which diameter type the name refers, most probably geometrical diameter.

*P4L107: "modelled" might cause ambiguity. Consider another word, e.g., mimic.*

**changed accordingly**

*P4L112–115: Is the concentration factor (=3000?) stable? What was the upstream particle number concentration to the inlet of the instrument? What were the particle number and mass concentrations during the measurement time?*

The real concentration factor is most probably lower than 3000, but difficult to measure for an ambient air ensemble, since the complete sampling line as well as the particle inlet have to be modified for a comparison.

**changes made to the manuscript:**
- **To provide an estimate on the concentration, we added two size distributions in the Supplemental Fig. S1, one with concentrator and one without. These were measured on ambient aerosols at the same site, but at a different day.**
- **Moreover, we added mean PM2.5 and PM1 mass concentrations to the manuscript. These were measured at the meteorological station with standard instrumentation during our ambient air experiment.**

*P5L120: Table1 Change "x" to multiplication sign "×". Change "…" to "to" or "−" and throughout the manuscript.*

**corrected**

*P5L127: Aren't those accumulated spectra rather than averaged ones? I am confused here.*

**changed "averaged" to accumulated**

P6L132: "much stronger Fe-signals". Please quantify such enhancement. How much % stronger?

The enhancement is quantified a bit later in the text. We would prefer to guide the reader's attention to the spectrum here.

*P6L141: In the histogram plots of Fig.1, what is the difference between the light red and dark red? Please add descriptions in the figure caption as well. Similar comment is applied to Fig. 3.*

The bars are transparent, and the dark areas are the overlapping areas. We prepared high-quality figures for publication, where these differences will be clearly visible.

P6L147: This is the first time to mention supporting information, thus it should be Fig. S1 rather than S4. Please revise accordingly throughout the manuscript.

**Corrected. We re-organized the Supplement.**

*P7L151: Such signal enhancement of 4 is compared to which value at which wavelength? As shown in Fig2, when compared to the lowest value, they are 7 for diesel soot and 4 for mineral dust; comparing the wavelength at 248. 3 and ~247 nm, an enhancement of Fe-54 signal is >2 and <2 for diesel soot and Arizona test dust, respectively.*

Because a clear baseline for a minimum yield does not exist (e.g. because also resonances with other lines occur, see Fig. 2), the effect can hardly be quantified with the few wavelength points we could address in these difficult and time-consuming experiments. We therefore estimated a conservative value of 3-4 from the lowest points to the maxima in Fig. 2.

*P7L155–160: Change this sentence to "Thomson et al. (1997) observed ....bulk material." Similarly, please revise the sentence "Wade et al…".*

**corrected**

*P7L158: Give the full name of "MALDI".*

**corrected**

*P7L160: Add reference.*

We cannot identify the issue. There are three references for this statement.

*P7L164: Fig. 2 please use the same scale for OPO wavelength in panel (a) and (b). Add Y axis with the corresponding units for atomic absorption spectrum and for KrF laser spectrum.*

**changes made to the manuscript:**
- **Fig. 2 is a complete new version showing new data and with the same OPO wavelength scale.**
- **The values measured with the spectrometer are in arbitrary units, as now indicated in the figure caption, and NIST data are only relative values in a limited interval (described in the referenced literature).**

*P7L170–172: Move to introduction.*

These publications are also referred in the introduction. We believe that this comparison to the literature is essential for the discussion here.

*P7L178: Give the full name of "RIMS". 4*

The full name was previously given in Line 174.

*P7L183–187: Move to introduction.*

**changed accordingly**

*P8L192–193: Please keep consistency, either use % or mg/kg for mass fraction.*

**corrected**

*P10L244: If refer to organics, m/z 39+ should be C3H3+.*

**corrected**

*P10L245–252: Move the data analysis part to the method section.*

**changed accordingly**

P11L259: Add more references for TMA, e.g., Angelino et al., 2001; Pratt et al., 2009; Köllner et al., 2017.

**Köllner et al., 2017 added**

*P11L269–270: What it the threshold of Fe signal(s) to identify Fe-containing particles.*

After resampling and baseline correction of the raw data, a peakfinder algorithm from the bioinformatics toolbox of matlab was used to identify peaks. The prominence threshold value was 1.5, which was determined in extensive tests on many individual particles by hand. Subsequently, 50% of the single spectra showing $^{54}Fe$ in the algorithm (black bars in Fig. 5) were manually cross-checked. Consequently, we are very confidential that we correctly identified the Fe-containing particles with virtually no false positive results.

**changes made to the manuscript:**
- **We added a statement on the manual cross-check to the text.**

*P12L290–292: Fe can be detected in different particle types does not suggest less particle matrix effect. Please see the first major comment and rephrase the sentence, as well as the following statement.*

**changes made to the manuscript:**
- **As previously discussed, we rephrased this sentence according to the reviewer's comments.**
  "This suggests that the resonant ionization allows a more universal and secure detection of Fe."

*Supplement Figure S4: Can you really detect particles in the size range from 0 to 200 nm? What is the detection limit of this instrument? Please add such information in the method section in the main manuscript, and only show the data within the detection limits.*

Similar to other instrumental realizations using large, ellipsoidal mirrors guiding scattered light to the photomultipliers, such as ATOF-MS (Su et al., 2004) and SPLAT (Zelenyuk et al., 2005), the detection efficiency drops below about 250 nm and it approaches zero around 100 nm. The efficiencies and limits of our instrument are published in the referred literature (Li et al., 2011)

**changes made to the manuscript:**
- **We changed the size distribution plots according to the suggestions of the reviewer and removed the particle counts up to 200 nm size, which may result from artefacts.**

Response to Anonymous Reviewer #2
We thank the reviewer for the valuable comments. Please see our reply below.

Note:
*Reviewer comments are in italics.*
Author responses are in normal format.
**Changes** that were made to the manuscript are in **bold** face with examples of the new text in blue.

*"[...]With the acceptation of some minor improvements in the reporting of the results, the technical merits of the manuscript is good. However there are some major concerns about the atmospheric relevance of the field study presented. Whilst this field data supports the technical development, it does not offer insight into the atmospheric implications of the presence of Fe in the environment in which it was measured. As this is a key requirement of this journal, the authors are requested to provide this discussion or consider submitting to a technical journal such as AMT."*

With our study, we aim on a broad scientific audience including scientists who are more focused on interactions between the atmosphere, the oceans and the biosphere than on specialized measurement technologies. To draw their attention to the application potential of our study, e.g. for existing and future interdisciplinary projects on micronutrient cycles, we aim on the publication in ACP as an open-access journal with large outreach. However, the data shown in our study already provide interesting insights into atmospheric iron transport pathways. Therefore, we extended the discussion of our data and its implications for previous as well as future studies on metal transport.

**changes made to the manuscript:**

- **We added a new part discussing the mixing state of Fe-containing particles, compared it to previous studies and discussed implications for future studies.**
  "Although our field study provides only a limited dataset, some general implications can already be derived. The internal mixing of Fe with sulfate or organic acids is assumed to be crucial for Fe dissolution, and thus for the anthropogenic increase of bioavailable iron input to the oceans (Li et al., 2017). Previous studies indicated that the Fe transport into the sea is dominated by coal combustion particles containing sulfate in Asia (Furutani et al., 2011; Moffet et al., 2012), while the majority of Fe-containing particles in Europe are mixed with nitrate and were attributed to traffic activities (Dall'Osto et al., 2016). Similar to our experiment, these studies found strong internal mixing of many Fe-containing particles, such as biomass burning signals with coal combustion contributions in Asia and secondary nitrate with Fe in Europe. However, Fe-particles with sea-salt signatures were negligible in the SPMS studies and mixtures of Fe and OC were a minor fraction (Furutani et al., 2011; Dall'Osto et al., 2016). In our study, these particles were the most abundant types of Fe-containing particles, if resonant ionization was applied, see Fig 5(b), while for non-resonant ionization, particles with EC signatures were dominant (Fig. 5(a)). Taking into account that the aforementioned SPMS studies utilized non-resonant LDI of Fe at 266 nm, Fe-transport in organic and salt/mixed aerosols might have been underestimated. Electron microscopy studies of individual particles in Asia frequently revealed thick coatings of secondary compounds and organic matter around Fe-rich particle components (Li et al., 2017; Moffet et al., 2012). Ultra-fine Fe-containing particles, such as soot from traffic emissions, can enter the long-range transported accumulation mode via agglomeration with larger particles and condensation of organic vapours, secondary nitrate or sulfate. In our study, we observed a high prevalence of Fe in sea salt and OC particle types, indicating the importance of these pathways for transport of biologically relevant Fe."

- **The conclusion section was extended to better address the general implications for SPMS applications in atmospheric chemistry.**
  "Not all physical details are fully understood, and the signal enhancement effects providing the basis for the improved metal detection efficiency are difficult to quantify for the different particle types. However, our results show that the increase in sensitivity is moderate for particular Fe-rich aerosols, such as the about 3-fold signal enhancement for Arizona test dust. The resonance enhancement appears to become more effective for mixed particles with smaller Fe-contributions such as in the ambient air experiment, where about 10 times more particles revealed Fe-signatures in direct comparison with non-resonant ionization. Taking into account the lower hit rate of the KrF laser, a downside related to its lower photon energy, the overall efficiency to identify Fe signatures in a single-particle mass spectrum was increased by a factor of about 20 in our ambient air study.
  The coincidental matching of the KrF laser line with a strong absorption of Fe atoms allows an easy and straightforward application of the resonance effect in the field. For direct comparison of KrF with ArF lasers, it has to be considered that the lower photon energy of KrF laser is associated with a reduced hit

rate and different mass spectral signatures of other particle components, e.g. organics. Further studies are required to evaluate these differences. Of note, because of its rather high pulse energy and the flat-top beam profile, the hit rate of the KrF laser was about 40−50 % in our experiment, which is still more than the values that are typically achieved with the most common laser line in SPMS, the Nd:YAG at 266 nm. Exploiting the resonance effect for other metals than Fe requires a tuneable Nd:YAG-OPO system, which is, however, more difficult to operate."

*Minor Comments:*

*L51 Most of the iron in mineral dust is in the form Fe2O3 (hematite) which is also insoluble. A distinction between soluble and insoluble Fe should be made as it affects bioavailiblity which is a major justification for carrying out this work.*

We rephrased the respective sentence to make the distinction clearer.
- **changes made to the manuscript:**
  "The highly soluble, and thus more bioavailable Fe from anthropogenic aerosols that adds to the larger flux of rather insoluble mineral dust…"

*L59 Replace 'were' with 'have been'.*
**corrected**

*L61 Replace/remove 'herein', as it implies it is a feature if this study.*
**corrected**

*L63 Not all SPMS is bipolar, e.g. PALMS.*
**corrected - statement on both polarities removed**

*L64 Replace 'typical ionization products' with 'typical observed ions'*
**corrected**

*L65 It would be useful to be more specific about which metals have been detected. Provide some references.*
- **We added a new sentence with examples and references.**
  "For example, vanadium can indicate ship emissions (Healy et al., 2009; Ault et al., 2010) and signal patterns of e.g. aluminium, silicon and calcium point on soil dust particles (Sullivan et al., 2007)."

*L68 The matrix effect will influence all compounds, not just the minor ones. The description of the matrix effect should be developed further.*
- **We rephrased the sentence and provided a clear link between the matrix effects and its physical origins that are provided in the following sentence.**
  "For example, the particle's humidity and its main composition can have a strong effect on the detection of particle compounds (Neubauer et al., 1998), known as matrix effects. These effects are associated with several poorly determined interactions at the particle surface and in the desorbed plume affect ion formation (Reilly et al., 2000; Reinard and Johnston, 2008; Hinz and Spengler, 2007; Murphy, 2007; Wade et al., 2008; Hatch et al., 2014; Schoolcraft et al., 2000), reduce detection efficiencies and complicate quantification approaches (Healy et al., 2013; Gemayel et al., 2017; Gross et al., 2000; Fergenson et al., 2001; Qin et al., 2006; Zhou et al., 2016; Shen et al., 2019)"
- **Furthermore, we specified our statements on matrix effects and removed this term from the discussion of our results, as suggested by reviewer 1.**

*L75 The description of LDI and REMPI are a little conflated. It would be easier to follow if they were described in separate paragraphs. It should also be clear when talking about the techniques generally, or specifically to SPMS. The reference Gunzer et al (2019) is a review paper not specific to SPMS for example.*
- **We rephrased the sentence to be more specific here.**
  "In such a two-step approach, thermal or laser desorption (LD) is often followed by Resonance-Enhanced Multiphoton Ionization (REMPI), a gas-phase ionization technique that is highly sensitive and selective for aromatic molecules (Gunzer et al., 2019) The LD-REMPI approach yields detailed mass spectra…"

*L87 The conclusions of the present study and those of Schade et al (2019) should be clearly separated.*
**changes made to the manuscript:**
- **We removed this sentence and added the statement on the combination of both technologies in the conclusions section.**

*Experimental Section:*
*An introductory paragraph at the start of this section would be useful to set the scene for the experimental approached that follow. A better description of the instrument geometry should be provided. The authors reference an early technical paper for the Hexin instrument (Li et al 2011) but it is unclear what other modification have been made (if any) that could influence the instrument performance. Are any prior publication made with this instrument platform (e.g. Schade et al 2019)?*

**changes made to the manuscript:**
- **We re-structured the Experimental section (now "Methods") and introduced sub-headings, added the requested technical information and further references.**
**example on new text on the basic instrumental design:**
"Briefly, its instrumental layout is conceptually close to the ATOF-MS (Su et al., 2004) with an aerodynamic lens inlet and an optical sizing unit that comprises of a pair of 75 mW cw-lasers at a wavelength of 532 nm, ellipsoidal mirrors and photomultipliers. The dual-polarity mass spectrometer is designed in Z-TOF geometry, as introduced by (Pratt et al., 2009). For further details, e.g. the inlet particle transmission and detection efficiency, we refer to the literature (Li et al., 2011; Zhou et al., 2016). After the laboratory experiments, we implemented delayed ion extraction ($\Delta t=0.4$ µs) using high-voltage switches (HTS31-03-GSM, Behlke GmbH, Germany) to improve the peak quality in the ambient air experiments (Vera et al., 2005; Li et al., 2018)"

*Results and Discussion:*
*A short introductory paragraph at the start of this section would also be helpful. The subsection titles should have some equivalence e.g. 3.2 Resonance enhancement of trace metals.*

**changes made to the manuscript:**
- **We changed the subsection headings accordingly.**

*In the lab experiments, why were different OPO wavelengths used for the soot (250.0nm) and test dust (242.2nm)?*

We thank the reviewer for his attention. We corrected the figure and text accordingly, that both for the soot and the dust the same wavelengths are compared. Now, in all spectra, the reference wavelength is shorter, i.e. the photon energy is higher. Thus, possible effects of a higher photon energy, e.g. photoionization thresholds, are excluded and the enhancement can be attributed to the resonance with a higher level of certainty. Only for Zn, the reference wavelength is longer, which is associated with the instrument's UV limit and explained in the text.

**changes made to the manuscript:**
- **new Figure 1 now shows a consistent wavelength comparison for Diesel soot and Arizona dust.**
- **statement on the exception of a higher wavelength for Zn added in the text.**
- **new data shown in Supplemental Figure S5 presents three-wavelength comparison for all Fe, Mn and Zn in NIST urban dust.**

*Figure1 What is the right hand axis of panel (B)?*
**changes made to the manuscript:**
- **we removed this artefact in the new version of Figure 1.**

*L152 'remarkably width'*
**corrected**

*L182-186 Seems to be introductory material. Could this be moved to the introduction?*
**moved to the introduction, according to the reviewer's suggestion.**

*L190 The NIST reference material is described as 'well characterised'. A summary of this characterisation should be given e.g. from NIST certificate of analysis https://www-s.nist.gov/srmors/certificates/1649b.pdf*
**We added the direct reference to the certificate in the methods section.**

*L200 'prove that not only the sum signals of the metals are higher in resonant case, but also more individual particles reveal their signatures.' This sentence is unclear and should be re-written with reference to limit of detection.*

We rephrased the sentence. An absolute value for a limit of detection can unfortunately not be provided, but the discussion on the enhancement's magnitude was strengthened in the discussion and the conclusion.

**changes made to the manuscript:**

- **sentence rephrased**

  "The histogram plots (d…f) prove that the enhancement does not only result from a minority of particles that contribute especially strong ion signals. In contrary, a higher number of individual particles reveal signatures of the respective metals, which indicates a more secure and sensitive detection."

- **magnitude of enhancement discussed, e.g.**

  " …the improved metal detection efficiency are difficult to quantify for the different particle types. However, our results indicate that the increase in sensitivity is rather moderate for particular Fe-rich aerosols, such as the about 3-fold signal enhancement for Arizona test dust. The resonance enhancement appears to become more effective for mixed particles with smaller Fe-contributions like in the ambient air experiment, where about 10 times more particles revealed Fe-signatures in direct comparison with non-resonant ionization."

L213 Data that describes this instability should be provided.

- **information provided**

  In our experiments, thermal lensing problems of the irregularly triggered OPO system reduced its pulse power and stability, resulting in a shot-to-shot variability of the pulse power up to about 30 % .

*Figure 4 Caption 'The two lasers fired alternately on 1500 particle each.'*
**corrected**

*Conclusion: Some quantitative estimation of the enhancement achieved should be given. The trade-off between metal enhancement and lower particle detection rates should be highlighted.*

**changes made to the manuscript:**

- **magnitude of enhancement discussed:**

[revised manuscript text omitted]